# Implementation of SLAM-Based Online Mapping and Autonomous Trajectory Execution in Software and Hardware on the Research Platform Nimbulus-e

**DOI:** 10.3390/s25154830

**Published:** 2025-08-06

**Authors:** Thomas Schmitz, Marcel Mayer, Theo Nonnenmacher, Matthias Schmitz

**Affiliations:** 1Institute of Vehicle Systems Engineering, Ulm Technical University of Applied Sciences, Prittwitzstraße 10, 89075 Ulm, Germany; 2Institute of Medical Technology and Mechatronics, Ulm Technical University of Applied Sciences, Prittwitzstraße 10, 89075 Ulm, Germany; marcel.mayer@thu.de; 3Computational Science and Engineering, University of Ulm, 89081 Ulm, Germany

**Keywords:** individual corner modules, SLAM, autonomous vehicles, trajectory planning, Nimbulus-e, online mapping, nonholonomic constraints

## Abstract

This paper presents the design and implementation of a SLAM-based online mapping and autonomous trajectory execution system for the Nimbulus-e, a concept vehicle designed for agile maneuvering in confined spaces. The Nimbulus-e uses individual steer-by-wire corner modules with in-wheel motors at all four corners. The associated eight joint variables serve as control inputs, allowing precise trajectory following. These control inputs can be derived from the vehicle’s trajectory using nonholonomic constraints. A LiDAR sensor is used to map the environment and detect obstacles. The system processes LiDAR data in real time, continuously updating the environment map and enabling localization within the environment. The inclusion of vehicle odometry data significantly reduces computation time and improves accuracy compared to a purely visual approach. The A* and Hybrid A* algorithms are used for trajectory planning and optimization, ensuring smooth vehicle movement. The implementation is validated through both full vehicle simulations using an ADAMS Car—MATLAB co-simulation and a scaled physical prototype, demonstrating the effectiveness of the system in navigating complex environments. This work contributes to the field of autonomous systems by demonstrating the potential of combining advanced sensor technologies with innovative control algorithms to achieve reliable and efficient navigation. Future developments will focus on improving the robustness of the system by implementing a robust closed-loop controller and exploring additional applications in dense urban traffic and agricultural operations.

## 1. Introduction

The primary goal of this project was to develop a nimble autonomous vehicle platform capable of mapping its environment, planning a trajectory, and following it autonomously. Dense urban city traffic and agricultural operations in confined spaces are typical applications of vehicles that can maneuver effectively under the given geometric constraints. The Institute of Vehicle Systems Engineering at Ulm University of Applied Sciences is currently developing the autonomous concept vehicle Nimbulus-e in the strategic field “Intelligent Commercial Vehicles” with the aim of maneuvering as agilely as possible in confined spaces. Autonomous vehicles promise enhanced safety, efficiency, and comfort in transportation. To achieve the targeted high agility under the conditions mentioned, large road wheel steering angles are necessary to reduce the turning radius. This is accomplished by individual steer-by-wire corner modules with in-wheel motors on all four wheels. With this architecture, the vehicle is able to follow arbitrary trajectories. These trajectories can be derived from maps of the environment based on path planning algorithms. The Nimbulus-e is equipped with various sensors to collect the environmental information and for vehicle motion control. The objectives included integrating hardware components such as LiDAR sensors, actuators, controllers, and implementing software algorithms for SLAM and trajectory planning/optimization. In the first step, open loop control is used for the vehicle motion. The verification of the approach is demonstrated using full vehicle simulation with ADAMS Car (version 2024.2) and a scaled physical prototype vehicle. In a later step, the open loop-based predictions will be used as the baseline for a robust closed loop controller. Potential applications of the presented platform are in the field of agricultural robots, e.g., for crop protection purposes and nimble passenger transport.

In [1,2], the development of the autonomous research vehicle elWObot for maintenance work in fruit growing and viticulture is described. The vehicle has two independently steerable chassis modules with limited steering angles on both the front and rear axle. Vertical suspension travel is generated exclusively by the tire’s sidewall. The research paper [3] discusses the navigation of the elWObot based on environmental maps. A binary occupancy map is created for path planning, and the probabilistic roadmap (PRM) technique is used.

As part of the UNICARagil project [4], leading German research institutions have joined forces to develop innovative approaches for the mobility of tomorrow. In this project, independent corner modules with wheel hub motors are used as well. A wheel steering angle of 90° allows the vehicle to maneuver agilely. The platform is scalable and can be equipped with different cabins for individual applications. The vehicle uses an inertial measurement unit (IMU), a LiDAR, two Radar sensors, and cameras [5]. An approach for designing the motion controller based on a given target trajectory is described in [6].

As a technology platform for urban mobility of the future, Schaeffler developed the People Mover, which was unveiled to the public for the first time at the 2018 IAA in Hanover. This four-seater vehicle features four individual corner modules with wheel hub motors and steer-by-wire technology as well. It is both electric and autonomous [7,8].

Other approaches to the development of mobile work platforms and agile autonomous vehicle architectures worth mentioning here include Amazon Zoox [9], Naio TED agricultural robot [10], Rinspeed Snap [11], Vitibot Bakus [12], and Yanmar Smash [13].

The main focus of this paper is to describe the full scope of the Nimbulus-e project, i.e., the development of a new, innovative, nimble platform concept, using well-established methods in terms of motion control, mapping, and localization. From the educational point of view, the project has been initiated to strengthen the skills of engineering students in the fields of special vehicle architecture design and vehicle motion control. So far, about 20 Bachelor- and Master projects have been completed. This paper summarizes the current status of the project.

## 2. Materials and Methods

### 2.1. Systems Engineering-Based Target Setting

As is customary in the automotive industry, the development of the Nimbulus-e was based on the systems engineering approach [14,15], whereby requirements are first defined at the vehicle level and then specified at the system level and finally at the component level. The starting point for the project was the idea to develop an agile autonomous multipurpose platform, which can effectively maneuver in confined areas. Based on this objective, the following workstreams have been identified and applied to this project:Development of a suitable vehicle underbody architecture including actuators supporting the agility objective,Development of a method for a real time mapping of the environment during operation including selection of suitable sensors,Development of a method for generating and optimizing the trajectory based on the environmental map,Open-loop autonomous control of the vehicle within a known static environment.

### 2.2. Vehicle Architecture

Considering the cost and resource situation in the university sector, it was decided to start with the design and development of a scaled prototype in the first place. A scale of 1:3.5 was chosen, which enabled using inexpensive hardware partially manufactured in the university’s own workshops. The maximum voltage level could be kept at 36V, which allows working without special precautions. The following sections will describe the mechanical and electrical hardware used for the scaled prototype. The full vehicle is shown in Figure 1.

#### 2.2.1. Suspension Design

In order to meet the agility objective of the vehicle, the turning radius of the vehicle has to be minimized. Conventional vehicles with front wheel Ackermann steering only are compromised with regard to the achievable turning radius. With individual corner modules for all four wheels, this problem can be resolved [8,16]. Figure 2 shows the suspension design chosen for the Nimbulus-e prototype and the kinematical model in ADAMS Car.

The special characteristic of this setup compared to conventional solutions is that the entire axle is connected to the body at just one interface point A. This point is also the pivot point of the axle and allows a 360° rotation by means of a steer-by-wire actuator which is mounted between fork and vehicle body. For the prototype, the steering angles are limited to about ±90° to avoid overarticulation of the connecting cables. A traditional double wishbone suspension is attached to the fork to allow vertical movement of the wheel. A strut comprising a spring and a shock absorber is mounted between the fork and the lower control arm. All four wheels are driven by wheel hub motors. Hence, the total vehicle can be controlled by eight control inputs. With such a suspension architecture, it is not only possible to reduce the turning circle, but it is also possible to maneuver to any position in the 
xy
-plane with arbitrary orientation of the vehicle body as shown in Figure 3.

As already shown for a steerable five-link suspension in [16], a caster angle of 0° has been chosen for the Nimbulus-e as well. In this configuration, the aligning torques resulting from the tire/road contact are purely due to the tire’s pneumatic trail. A self-locking worm gearbox has been selected for the steering system. With a ratio of 18:1, the gear matches the steering motor torque to the tire forces. The advantages of this concept are that the alignment torque is counteracted by the body structure rather than the steering motor and that common components can be used for all four corner modules, reducing piece and tooling costs. However, more friction is introduced into the steering system. The large vertical lever arm between the tire patch and the pivot point *A* drives the need for a robust bearing for the fork, to enable a stiff load transfer between suspension and body. The tapered roller bearings are therefore installed in the so-called “O-arrangement” (Figure 4). Both the steering motor bracket and the gearbox housing were produced using FDM (Fused Deposition Molding) 3D printing. FDM offers significant advantages such as material diversity, cost efficiency, ease of use, fast production times, and design flexibility, making it a popular choice for rapid prototyping.

#### 2.2.2. Body Structure

The frame of the vehicle consists of several extruded aluminum square tubes with an edge length of 25mm and a thickness of 2mm that are bolted together. The required rigidity of the construction is achieved using a node in which three interlocking square tubes are bolted together using three bolted connections (see Figure 5). This design was derived from the principle of the armored barrier, which was invented during the Second World War and makes it possible to create very torsionally rigid structures from very little material. The company XYZ CARGO (Hamburg, Germany) offers build-your-own cargo bikes using this technology. It is well suited for prototypes since the manufacturing process is simple and the components are inexpensive.

#### 2.2.3. Electrical Architecture

Figure 6 shows the arrangement of the electrical components in the Nimbulus-e. All four wheels have an integrated wheel hub motor. The system is steered by one stepper motor at each corner. A plastic housing from FIBOX (Porta Westfalica, Germany) above each corner module houses the control units for the steering and wheel hub motors and protects the electronic components from external influences. A fan motor (Sunon, Kaohsiung City, Taiwan) is fitted on the side of each controller box to ensure that the components are well ventilated, and each corner module is equipped with a steering angle sensor (ETA25K-series, Megatron, Putzbrunn, Germany). In addition, two end position switches (ME8108-series, Moujen, Morgan Hill, CA, USA) are installed on each wheel suspension to avoid overarticulation. The main box is located at the rear center of the vehicle. This plastix box houses two Arduino Mega micro-controllers (Arduino, Monza, Italy) which are responsible for controlling the vehicle motion. There is also a fuse holder, a fan motor, an EMC bus bar, a radio-controlled emergency stop, and a central power strip. The main PC is connected to the Arduino A5.1 via the USB interface. Underneath the main box are two bicycle batteries that power the electrical system. The voltage converters on the bottom of the main box convert the 36 V DC voltage of the batteries to 12 V DC, 230 V AC and 5 V DC. For safety reasons, there is a main switch in the rear of the vehicle that can be used to cut or release the battery voltage. An emergency stop switch is located next to the main switch.

As stated above, two Arduino Mega micro-controllers are used for vehicle motion control. Both Arduinos communicate via the serial tx-rx-pins present on the boards. Arduino A5.1 is the main controller and is responsible for the communication with MATLAB (version R2024b), the wheel hub motor controllers, and reads all sensor data. The Arduino A5.2 controls only the stepper motors and reads the steering angles so that the Arduino A5.1 has minimal delay from steering.

The stepper motor actuates the steering depending on the control signal it receives from the Arduino A5.2. The S-Series NEMA 24 closed-loop stepper motor (OMC Corporation Limited, Jiangning, China) is a powerful stepper motor equipped with an integrated 1000 RPR (4000 CPR) encoder to ensure precise position feedback and efficient control (Figure 7a). With a torque of 3.0 Nm, this motor is designed for demanding applications where high accuracy and stable performance are critical. In standard full-step operation mode, the step width is 1.8° (200 steps per revolution). In microstep operation, smaller steps can be realized. Two cables A and B with two different plug connections lead out of the stepper motor. The motor extension cable B has a 4-pin round connector. The related four poles form the two pole pairs of the motor, which set the motor in motion when energized. The encoder extension cable A provides the controller with feedback via the encoder about the current position of the shaft in order to avoid step losses.

The stepper motor controller used is the CL57T(V4.0) closed-loop motor controller for NEMA 17, 23, and 24 stepper motors. It operates with a voltage of 24 V to 48 V and offers an adjustable output current of 0 A to 8 A, which is suitable for demanding applications. The number of steps per full rotation can be changed via the pin configuration. Due to the existing worm gear between the stepper motor and steering axle with a ratio of 18:1, the largest possible step width was selected, corresponding to 200 steps of the stepper motor for one revolution of the motor output shaft. With this ratio, 10 steps correspond to one degree road wheel steering angle. In addition, all wheels are set so that a rotation with the direction pin on HIGH corresponds to a rotation of the wheel in the top view in a mathematically positive direction (counterclockwise).

The wheel hub motors of the Nimbulus-e are external rotor brushless DC (BLDC) motors. They are permanent magnet synchronous machines (PMSM). Synchronous machines are three-phase machines that are controlled by power electronics. Due to modern control methods, synchronous machines have become increasingly important in drive technology. The specific model selected is the “Vbest life 36 V 350 W hub motor for 8-inch electric scooters” (see Figure 7b).

Electronic speed controls (ESCs) provide the motor with the required voltage and also regulate the power of the motor. The name “VESC” is a collective term for motor controllers that work according to the Benjamin Vedder standard [17]. It is a non-commercial project where the application software “VESCTool” (version 6.05) is provided free of charge. The specific model selected for this application is FSESC 4.12 from “Flipsky”. The individual motors can be configured using the application software VESCTool. The wheel hub motors are controlled via a UART connection between the Arduino A5.1 and the master VESC controller, rear left (RL) with the ID0. All VESC controllers are connected to each other via a CAN bus (CAN Low and CAN High respectively). The controllers are interacted with from the Arduino A5.1 via their predefined IDs using the developed VescNimbulus class. The corresponding methods setRPS() and getRPS() have been developed to control and read out the respective rotational speeds of all four wheel hub motors.

From a functional point of view, the installed sensors used are divided into those used to control the actuators and those required for localization and mapping.

Steering angle sensors with a non-contact magnetic encoder from the “MEGATRON” ETA25K series have been selected. These sensors can detect steering angles of up to 360° and output a proportional voltage of 0 V to 5 V. The sensor signals are processed independently on both Arduinos and are calibrated individually for both micro-controllers, as the analog-to-digital converters (ADC) measure different voltages despite the same voltage being applied. In order to measure the wheel speeds and the position of the rotor in relation to the stator, there is a circuit board with three Hall sensors embedded in the stator iron cores (Figure 7b). The sensors output a voltage level between 0 V to 3.3 V, depending on their magnetic flux. These signals are used by the VESC for wheel speed control.

Key sensors include LiDAR for environment localization and mapping, ultrasonic sensors for obstacle detection, and a compass. These sensors provide the necessary data for SLAM and trajectory planning. The Ultra-Puck VLP-32C from Velodyne is a 3D scanning LiDAR sensor that achieves a horizontal field of view of 360° and a vertical field of view of 40°. A central built-in Adafruit BNO055 compass is used to determine the absolute orientation of the vehicle. This sensor is well suited for short and medium-term angle measurements, but may suffer from drift errors in long-term applications, which can be caused by the integration of gyroscope data. Although the magnetometer values help to correct this drift, they are sensitive to magnetic interference, which can lead to inaccuracies, especially indoors. A total of 16 HC-SR04 ultrasonic sensors are positioned on the vehicle. They ensure the widest possible coverage of the surroundings. The HC-SR04 ultrasonic sensor has a measuring range of 2 cm to 300 cm and a beam angle of ±15°. The ultrasonic sensors are used to detect obstacles in the immediate vicinity of the vehicle. Their positioning has been specifically coordinated so that the beam angles and the time-shifted transmission of the signals enable complete coverage of the surroundings and signal overlaps are avoided.

Comprehensive safety functions have been integrated to ensure the safety of the vehicle during operation. These include a radio emergency stop function that stops the vehicle immediately as soon as a signal is triggered with a remote control, as well as end stops that prevent the steering angles of the wheels from exceeding their specified limits of ±100°. The assigned voltage values are continuously monitored on interrupt-capable pins of the Arduino A5.1. As soon as an end stop is triggered, the radio emergency stop is pressed or the emergency button at the rear of the car is activated, the voltage value drops from 5 V to 0 V. The interrupt service routine (ISR) is called and the vehicle is transferred to the safe state.

## 3. Kinematical Model

Since the vehicle is operated at low speeds and low lateral accelerations, the movements of the wheels can be described by nonholonomic constraints [18]. Thus, the trajectory of the vehicle in the plane 
(x(t),y(t),ψ(t))
 can be converted into wheel steering angles and wheel speeds.

Nonholonomic constraints arise when additional restrictions on the system exist that originate not only in geometry but also in velocity. The classic example of a nonholonomic constraint is a wheel rolling without slip (Figure 8). On position level, it is clearly defined by the three tire patch coordinates 
$x_{R},y_{R},\delta_{R}$
, but is also restricted in the direction of the velocity, which can only run parallel to the longitudinal axis of the wheel without lateral slippage, as well as the rolling condition, which describes the relationship between translational velocity 
$v_{R}$
 and angular velocity 
$\omega_{R}$
 at a given wheel radius *R*. Nonholonomic constraints are often described by inequalities or differential equations. In the example of the wheel rolling without slip, the two nonholonomic constraints are: 
(1)
$$\begin{array}{ccc} \frac{v_{Ry}}{v_{Rx}} & = & \tan\delta_{R}, \end{array}$$

(2)
$$\begin{array}{ccc} v_{R} & = & \sqrt{v_{Rx}^{2}+v_{Ry}^{2}}=\omega_{R}\;\cdot \;R. \end{array}$$


As shown in Figure 9 the velocity vector of wheel center 
$v_{Ri}$
, 
i=1,…,4
 can be obtained from the forward kinematics as a function of the velocity of the vehicle center 
$v_{S}$
, the angular velocity of the vehicle body 
ω
, and the relative position in the vehicle 
$r_{S,i}$
 to
(3)
$$v_{Ri}=v_{S}+\omega \times r_{S,i}.$$

In components of the vehicle-fixed coordinate system 
$K_{F}\;$
, the velocity vectors of the wheels are thus
(4)
$$F_{{\underline{v}}_{Ri}}=\left[\begin{array}{c} v_{xi}\\ v_{yi}\\ 0 \end{array}\right]=\left[\begin{array}{c} v_{S_{x}}\\ v_{S_{y}}\\ 0 \end{array}\right]+\left[\begin{array}{c} 0\\ 0\\ \dot{\psi } \end{array}\right]\times \left[\begin{array}{c} \pm w_{b}/2\\ \pm t_{w}/2\\ 0 \end{array}\right]=\left[\begin{array}{c} v_{S_{x}}\pm \dot{\psi }\;\cdot \;t_{w}/2\\ v_{S_{y}}\pm \dot{\psi }\;\cdot \;w_{b}/2\\ 0 \end{array}\right]\;,\;i=1,\ldots ,4.$$

In order to control the planar vehicle motion with 
f=3
 degrees of freedom, the eight dependent control inputs 
$\delta_{Ri}$
 and 
$\omega_{Ri}$
 have to be determined as a function of the desired trajectory. This step is called forward kinematics and is fully described by the kinematic transformer shown in Figure 10. Using Equation (Equation 4), the wheel speeds 
$\omega_{Ri}$
 and wheel steering angles 
$\delta_{i}$
 can be unambiguously derived from the nonholonomic constraints (Equation 1) and (Equation 2). To determine the current position of vehicle’s center *S* from given steering angles 
$\delta_{Ri}$
 and wheel speeds 
$\omega_{Ri}$
 during operation, the equations must be evaluated in reverse order. This is known as inverse kinematics and in most cases can only be done by estimation. In the case of the Nimbulus-e, the number of input variables is greater than the number of unknowns. The conventional approach to solve this overdetermined set of equations is to minimize the sum of the squares of the errors. This is done using the so-called Moose–Penrose pseudoinverse. Estimating the position and orientation of a mobile robot from the data of its drive system is also referred to as odometry in robotics. Often, odometry is supplemented with additional sensors such as GPS or compass data to achieve more accurate results. This is shown in Section 4.1.

## 4. Mapping and Path Planning

The goal of autonomous navigation is to find an optimal path from a starting pose to a destination within a given environment. The environment may be static or dynamic. Once a map of the environment is available, an optimal path can be planned based on a defined cost function and specific constraints. The task of the vehicle motion controller is to follow the path as smoothly as possible. Due to the individually steerable wheels, the Nimbulus-e vehicle can follow almost arbitrary paths as long as the package constraints are met.

### 4.1. Map Generation Using SLAM

In the first step, a map of an existing environment will be generated. An Xbox controller based steering was implemented to control the vehicle for the creation of these maps. In order to allow the user to control the vehicle manually, a simple and intuitive operation of the vehicle is desirable due to the eight possible control variables. Performing both purely translational and purely rotational movements of the vehicle is possible. The vehicle can move linearly (all steering angles and speeds are identical) or turn on the spot. By means of this controller, the vehicle can be manually maneuvered through the environment in order to collect the information for the subsequent SLAM.

SLAM stands for Simultaneous Localization and Mapping. It is a method that addresses the problem of simultaneously creating a map of the environment and estimating the current position within that map. SLAM is used for autonomous navigation, including obstacle detection [19,20,21,22]. The utilization of SLAM by a mobile robot or vehicle is contingent on the availability of sensor data. Such data can be obtained from ultrasonic, LiDAR, or camera sensors. The Nimubulus-e, for instance, employs a LiDAR sensor to fulfill this function. A key challenge in SLAM is that in order to generate an accurate map, the robot must be accurately located; in order to determine its position, it must already have an accurate map. To address these challenges, various SLAM algorithms have been developed [23,24].

LiDAR continuously generates images of the environment. The current LiDAR-scan is processed by removing high points and ground, randomly reducing the total number of points, removing outliers, and performing a 3D-to-2D projection. Finally, the scan is passed to the selected SLAM object in MATLAB.

SLAM successively builds a map and tracks the driven trajectory by estimating the relative poses differences between two scans and searching for loop closures, i.e., comparing scans in the vicinity and estimating their relative pose difference. Both kinds of estimates are connected to a weight which is a measure of how good the estimation is. The resulting pose graph can be optimized and thanks to the loop closures possible errors are reduced, which would otherwise add up during the ride [25,26]. There exist two major implementations of SLAM in MATLAB. The lidarscanmap object from the MATLAB LiDAR Toolbox (version 2.3) and the lidarSLAM object from the MATLAB Navigation Toolbox (version 2.4). Both can estimate the local relative pose between adjacent scans purely from the visual 2D scan data or by additionally providing external pose estimates, in our case computed from odometry. The major differences are that the LiDAR Toolbox requires functions from the Navigation Toolbox for graph optimization after loop-closure detection, so the user has to call those functions themselves, the Navigation Toolbox encapsulates this nicely. Furthermore, the LiDAR Toolbox does solely rely on external pose estimates if provided, visual data is only used for loop closure detection, while the Navigation Toolbox uses the external pose estimates as a starting point for the visual search. Comparisons of the four versions can be found in Table 1.

To compute pose estimates, the four wheel speeds, the four steering angles, and the compass signal are available to determine the yaw angle and yaw rate. The velocity of the center of the vehicle can then be determined directly by rearranging Equation (Equation 4). Hence, 
$v_{S}$
 can be calculated from any wheel center velocity 
$v_{Ri}$
 derived from the measured wheel steering angles, the wheel speeds, and the yaw rate from the compass signal. Since the system is overdetermined, in this specific case, the median of the four results is used. Figure 11 shows the information that is exchanged and the time intervals at which data transmission occurs.

In the context of online SLAM, the scans are processed directly during the journey, with the map undergoing continual updates with each new scan. Achieving simultaneous processing of the data in real time necessitates the parallelization of the computing operations. The MATLAB Parallel Computing Toolbox (version 7.8) facilitates the execution of scan processing and SLAM computation in parallel with vehicle motion control.

Given the easier API, which simplifies parallelization, and the fact that visual data is not neglected during relative pose difference estimation of adjacent scans, the lidarSLAM object from the MATLAB Navigation Toolbox was chosen despite the better efficiency of the lidarscanmap object.

Figure 12 compares the results of different SLAM implementations with the ground truth and poses obtained from odometry. It can be seen that the implementation of the Navigation Toolbox is almost identical to the ground truth. This holds for both runs with and without supplied poses, as they are only used as initial guesses to speed up the computation. The LiDAR Toolbox also performs well without poses, although differences are present, e.g., at about 
(4.5,−1.5)
. The LiDAR Toolbox with poses is based solely on the pose estimates. As they are rather poor estimates, the difference is already large at the second scan (at about 
(4,0.2)
). For the later scans, loop-closures based on optical information partly correct the poses.

These results can be quantified by taking the root mean square error (RMSE) of each method compared to ground truth, with the results presented in Table 1. The Navigation Toolbox has a lower error by a factor of 3 and a factor of 8, concerning the calculation with and without provided pose estimates, respectively. Again, according to the highest RMSE, it is clear that the LiDAR Toolbox with pose estimates suffers significantly from inaccurate pose estimates. Furthermore, the visual approach of the LiDAR Toolbox yields worse results than the Navigation Toolbox; however, it is faster by 20%. The best method is the Navigation Toolbox with provided pose estimates, as this reduces both the error and the computational time, while using the approximately same amount of CPU as the version without pose estimates.

Figure 13 shows the created map of a SLAM run using the Navigation Toolbox and providing the pose estimates from odometry. The resulting map is a clear 2D image of the laboratory that includes the driven trajectory. From this map a two-dimensional occupancy map can be derived. This map comprises individual cells in a grid, each one indicating the probability of the cell being occupied. This type of map is called a probabilistic occupancy map. Values approaching 1 signify a high probability of obstruction within the cell, while values approaching 0 denote a high probability of the cell being obstacle free. Subsequent to the generation of the probabilistic occupancy map, a binary occupancy map is produced by applying a threshold which is subsequently used in the following chapter to calculate collision-free paths under given constraints. An example of a probabilistic and binary occupancy map is shown in Figure 14a,b, respectively, where the chosen threshold of 0.005 is based on Otsu’s method implemented by the MATLAB function graythresh().

### 4.2. Path Planning and Optimization

Based on the map created in Section 4.1, the desired path within the known environment can be determined. For this purpose, two different algorithms are used, both of which are implemented in the MATLAB Navigation Toolbox.

The Hybrid A* algorithm is a grid-based planning algorithm [27,28,29]. It generates a smooth trajectory in a given two-dimensional map for vehicles with nonholonomic conditions. The movement of the vehicle is described by the position of the vehicle center 
$x_{S},y_{S}$
 and the rotation around the vertical axis (yaw angle 
ψ
). Given the start and end poses, the algorithm calculates the shortest available route taking into account a specified minimum turning radius. The MATLAB function plannerHybridAStar was used for implementation. In the second step, the planned trajectory can be optimized with regard to certain parameters using the function optimizePathoptions in MATLAB. The smallest turning circle, the maximum speed, and maximum acceleration are taken into account. In addition, safety distances to obstacles representing the vehicle dimensions are specified. In order to consider the high maneuverability of the Nimbulus-e on the one hand, but also to achieve smooth trajectories on the other hand, the Hybrid A* algorithm was implemented taking into account a very small turning circle.

While nonholonomic constraints apply to the individual wheels of the Nimbulus-e, arbitrary trajectories can be generated for the vehicle center, i.e., the vehicle can turn on the spot or even follow a sharp rectangular trajectory. For this purpose, the so-called A* or similar algorithms can be used in general [30,31]. This algorithm was originally developed for vehicles with purely holonomic constraints. By selecting this algorithm, the full potential of the Nimbulus-e suspension system with regard to sharp traces can be utilized. The disadvantage is that the body orientation is not taken into account with A*. This means that a square vehicle base with the same side length as the Nimbulus-e vehicle length must be selected when planning the route to avoid collisions with the environment. Certain possible paths will be excluded in the first place as shown in Figure 15a. The full utilization of the capabilities of the Nimbulus-e suspension in terms of possible trajectories, including rotation at a standstill, can be achieved by selecting a very small turning circle in the Hybrid A* algorithm (Figure 15c). The disadvantage of Hybrid A* is that a pure parallel movement is not possible since the algorithm always tries to find a path in the direction of the longitudinal vehicle axis.

As an example, the path from a starting point **A** to a goal point **B** through a given static environment is planned using both the Hybrid A* and the A* algorithm as shown in Figure 16. While Hybrid A* defines a pose consisting of position and orientation to define the movement of the vehicle, A* does not consider orientation. The optimized trajectories, taking into account additional physical constraints, are shown in blue. Although the optimized trajectories resulting from the two algorithms appear similar, there are significant differences between the two approaches at the start and end points. To reach the end pose, the vehicle must turn and reverse into the parking space as shown in Figure 16a. This can only be achieved with Hybrid A* but not with A*, since the orientation of the vehicle after optimization is determined by the direction of the optimized trajectory. As explained above, with A* the individual length and width of the vehicle cannot be taken into account. The trajectory in Figure 16b was created taking into account the vehicle width (narrower dimension). If the vehicle length (larger dimension) was taken into account for the distances to the obstacles, the A* algorithm would not find a possible path. Since the Nimbulus-e has different dimensions in the longitudinal and transverse vehicle direction and requires consideration of the full pose, i.e., position and orientation, the Hybrid A* algorithm will be applied for the remainder of the project.

The advantages of the Nimbulus-e concept with individual corner modules in terms of maneuverability in confined spaces compared to a vehicle with traditional Ackermann steering are evident from the subsequent path planning simulation through a simple labyrinth.

Figure 17a shows that, enabled by its small turning circle in forward mode, the Nimbulus-e can move effectively and quickly from the starting point to the end point without complicated turning maneuvers. A vehicle of the same dimensions and Ackermann steering with a 40° wheel steering angle can only reach its destination with complex maneuvering and alternative forward and reverse driving (Figure 17b).

## 5. Results

### 5.1. Simulation with ADAMS CAR

An ADAMS CAR full vehicle model has been created for controller development. The dynamic behavior of the tires is modeled using the Magic Formula PAC2002 [32]. Compared to modeling the tire-road contact with nonholonomic constraints, this model takes into account longitudinal and lateral slip, which occur at higher longitudinal and lateral accelerations. Various friction conditions between tire and road can be considered as well. Since there are no dedicated test data available for the prototype wheels, surrogate test data had to be used for the ADAMS tire model. The open-loop controller, based on Equations (Equation 1), (Equation 2) and (Equation 4), has been implemented as a MATLAB block. The total system is analyzed using ADAMS CAR-MATLAB co-simulation (Figure 18).

The simulation is used to investigate how the system behavior changes with regard to vehicle speed and higher accelerations. In the first example, a circular target trajectory with a radius of 15 m has been defined as the reference path. The road surface is modeled as dry asphalt. For low velocities up to about 10 km/h, which is related to a lateral acceleration of 0.5 m/s^2^, the open-loop controlled trajectories align well with the target trajectory. The higher the speed, the higher the slip angles of the wheel, leading to a larger radius of curvature (Figure 19). In order to compare the simulation results with the physical vehicle behavior, a circular target trajectory was used as baseline.

As a metric for the position error, the relative deviation of the actual curvature radius compared to the target radius was analyzed for different velocities, i.e., lateral accelerations (Figure 20). For the ADAMS model, the deviation grows almost linearly with the lateral acceleration. For the physical vehicle test, no statistically significant error is evident for accelerations up to up to 0.5 m/s^2^. The model and simulation behave similarly between 0.5 and 2 m/s^2^. This aligns well with the intended working range of the Nimbulus-e vehicle. Above 2 m/s^2^, the test vehicle loses grip and slides extensively. This it due to the fact the prototype wheels on the workshop floor have a much lower grip level than the surrogate ADAMS tire model.

For the second simulation example, a reference path through a labyrinth derived from the Hybrid A* path planning algorithm is used. Figure 21 shows the desired and actual trajectories at different velocities. At low average velocities up to about 5 km/h, the two trajectories match well. As speed increases, the difference increases due to the higher slip angles on the tires. It can be concluded from the simulation results that the open-loop control based on nonholonomic constraints is a promising approach to predict the trajectory of the vehicle at low speeds in the vicinity of the actual vehicle position and will serve as an effective baseline for a closed-loop controller.

### 5.2. Autonomous Vehicle Control

The trajectories obtained in Section 4.2 are transformed into the corresponding values for the eight control inputs derived from Equations (Equation 1), (Equation 2) and (Equation 4). For autonomous driving of the vehicle, precise timing is mandatory, and time steps have to be consistent on all processors, i.e., MATLAB and the two Arduinos. To achieve a uniform time grid, the optimized path is interpolated at the desired time steps of 100 ms, providing the best compromise between accuracy and Arduino performance. Every 100 ms, the Arduino A5.1 reads the current odometry data (steering angles, wheel speeds, and yaw angle) and sends them to MATLAB which, in turn, triggers the transfer of the next line of control operators. The Arduino always stores one such line in advance. Thus, no huge storage on the Arduino is needed, but small delays in communication do not affect the operation. LiDAR scans are also stored every 100 ms. Ultimately, a comparison between different trajectories is possible, including the planned Hybrid A* trajectory, the optimized trajectory, the one obtained from inverse kinematics using odometry data, and the one achieved by locating the LiDAR scans on the map.

The actual starting point of the Nimbulus-e while testing is slightly adjusted since small wheel speeds cannot be controlled accurately with the given hardware (see Figure 22). Thus, the actual and the target trajectory align well in the beginning and differ continuously more with the traveled distance due to side slip and tolerances of the actuator outputs. The path derived from odometry data seems to be quite close to the path derived from LiDAR data, with errors increasing over time. Both are locally similar to the optimized trajectory. This is a fundamental finding for future work on closed-loop control.

The physical tests show a similar behavior as the ADAMS Car simulations, but the deviations over the distance driven are greater since more influencing tolerances and disturbances are evident.

## 6. Discussion and Outlook

As stated at the beginning of the paper, the focus of the Nimbulus-e project was to develop an innovative, autonomous, full-vehicle concept with high agility in confined spaces, using inexpensive hardware in collaboration with Bachelor’s and Master’s students. A unique vehicle architecture was selected with individually driven corner modules with steer-by-wire technology to achieve this objective. Due to the special wheel suspension architecture with road wheel angles up to 90° on all four corners the vehicle can follow almost arbitrary target trajectories. For path planning, it has been shown that the usage of the Hybrid A* algorithm is preferred over the A* algorithm since it covers vehicle dimensions and vehicle orientation. For localization and mapping well-established methods are applied using the respective MATLAB toolboxes. The lidarSLAM object from the MATLAB Navigation Toolbox was selected due to its advantages in terms of parallelization and accuracy. The initial application of the Nimbulus-e is to drive at low speeds and perform quasi-static maneuvers. For these kinds of maneuvers up to 2 m/s^2^, it is sufficient to model the wheel-to-road contact with nonholonomic constraints. The resolution of the steering system of 0.1°/step does not contribute to a significant error in this working range.

By means of the open loop controller developed based on nonholonomic constraints, target trajectories for low speeds in the vicinity of the vehicle’s current position can be predicted well. Environmental conditions such as friction between wheel and road, inclination of the road surface, asymmetries, inaccurate sensors, and actuators lead to deviations which can only be counteracted with a closed loop controller, which will be developed in the next step. This controller calculates the necessary corrections based on the deviation of the estimated trajectory from the target trajectory using the steering angles and wheel speeds predicted by the open-loop controller based on nonholonomic constraints.

There is a future need for applications of such platforms in the field of agricultural automation and flexible work machines. Mobile autonomous carrier platforms can carry flexibly interchangeable superstructures, for example, for leaf cutting, mulching, spraying, and garbage collection. However, it is always important that the systems are robust and economical from the user’s point of view. Another potential application is in the military sector, for example for supply logistics.

## Figures and Tables

**Figure 1 sensors-25-04830-f001:**
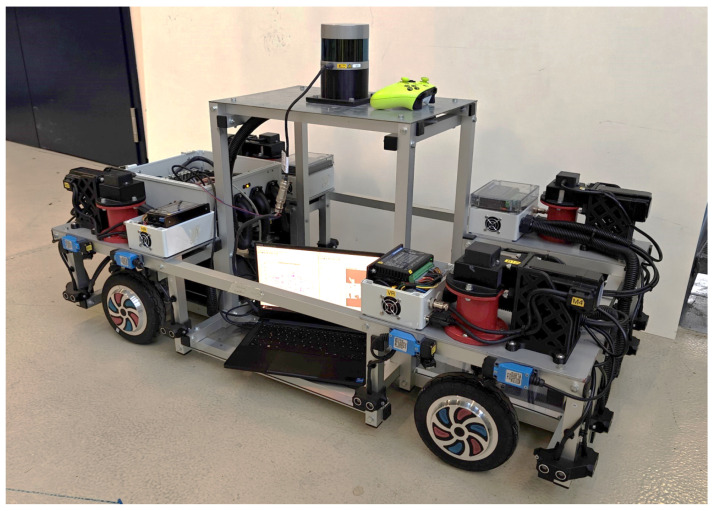
Nimbulus-e prototype vehicle.

**Figure 2 sensors-25-04830-f002:**
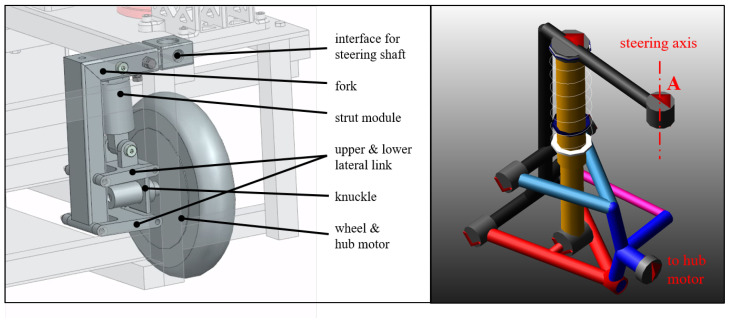
Nimbulus-e suspension design and kinematical model in ADAMS Car.

**Figure 3 sensors-25-04830-f003:**
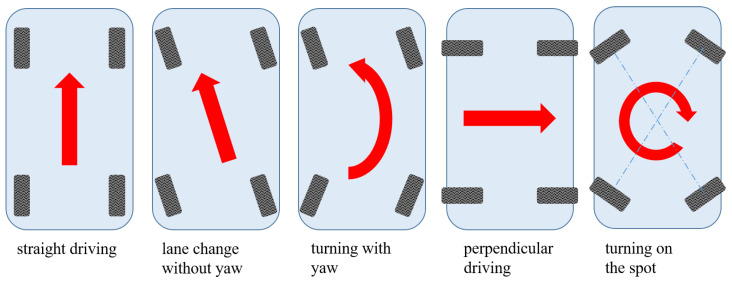
Maneuverability with all wheel steering.

**Figure 4 sensors-25-04830-f004:**
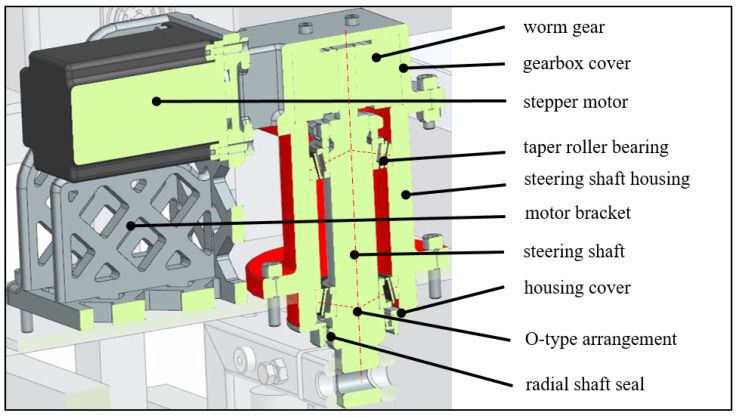
Steering system.

**Figure 5 sensors-25-04830-f005:**
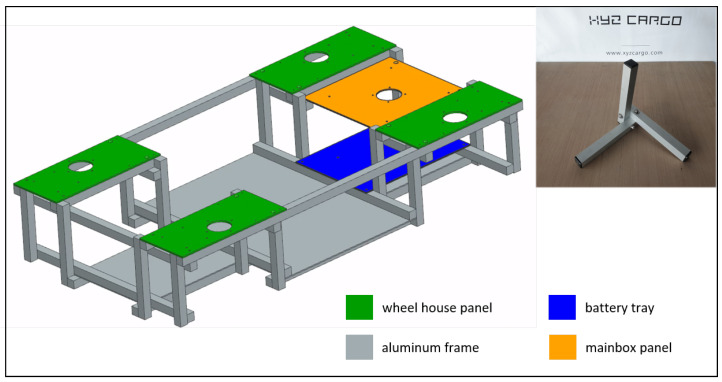
Nimbulus-e body structure.

**Figure 6 sensors-25-04830-f006:**
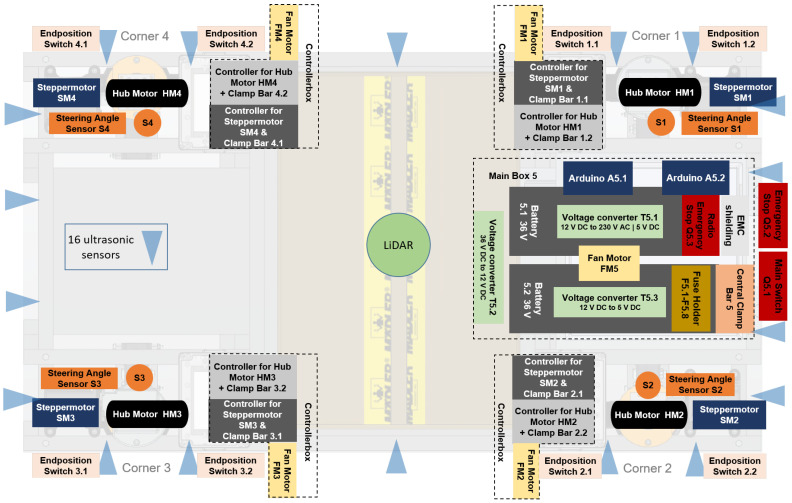
Schematic arrangement of the major electrical components in plan view.

**Figure 7 sensors-25-04830-f007:**
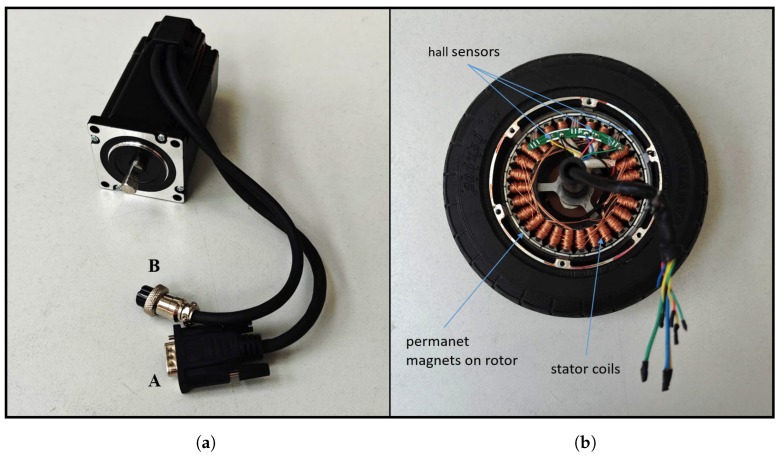
(**a**) S Series Nema 24 Steppermotor Stepperonline, (**b**) wheel hub machine Vbest life 36 V.

**Figure 8 sensors-25-04830-f008:**
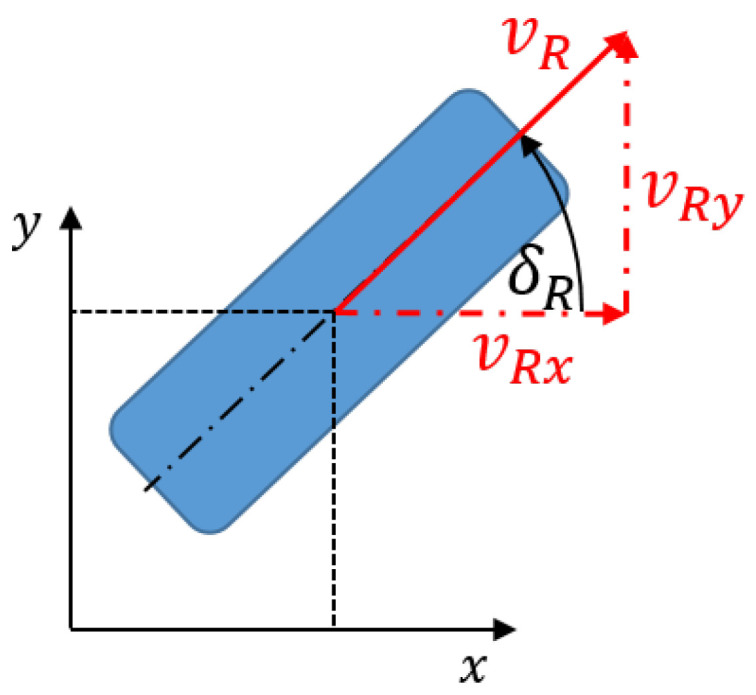
Wheel-to-road contact model.

**Figure 9 sensors-25-04830-f009:**
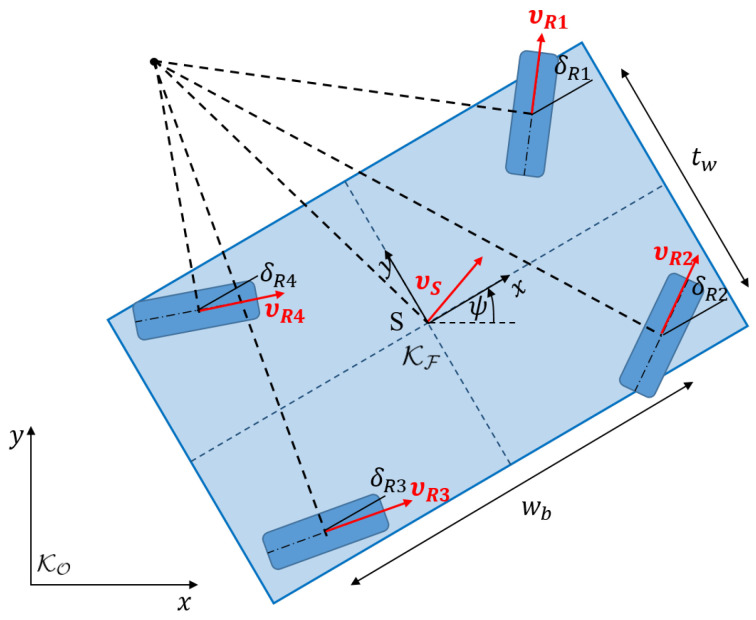
Plane vehicle model.

**Figure 10 sensors-25-04830-f010:**
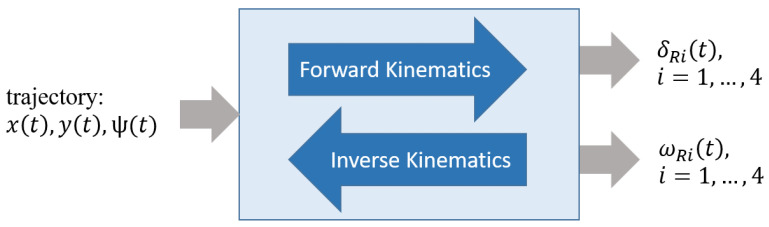
Forward and backward kinematics.

**Figure 11 sensors-25-04830-f011:**
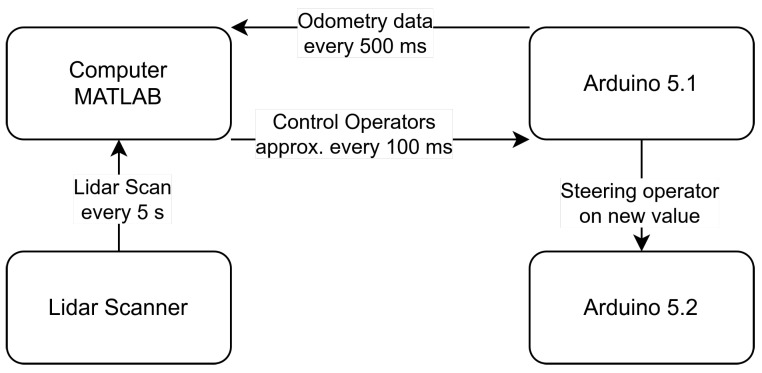
Communication and synchronization between systems.

**Figure 12 sensors-25-04830-f012:**
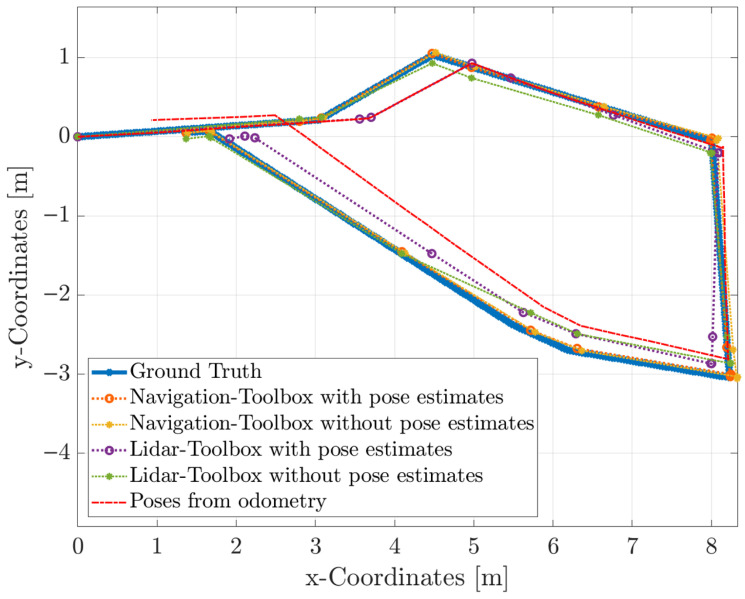
Comparison of trajectories derived from different SLAM methods.

**Figure 13 sensors-25-04830-f013:**
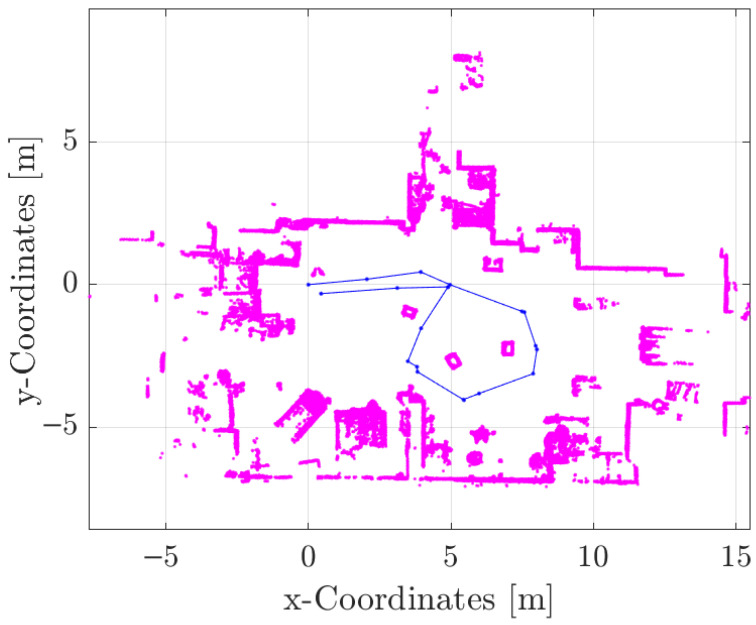
Mapped environment with obstacles (pink dots) and trajectory (blue line).

**Figure 14 sensors-25-04830-f014:**
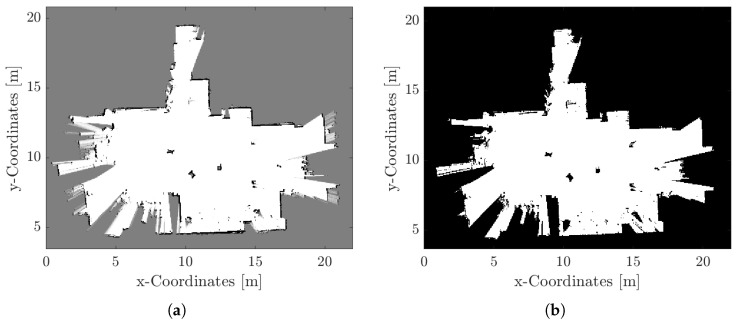
(**a**) Probabilistic and (**b**) binary occupancy map.

**Figure 15 sensors-25-04830-f015:**
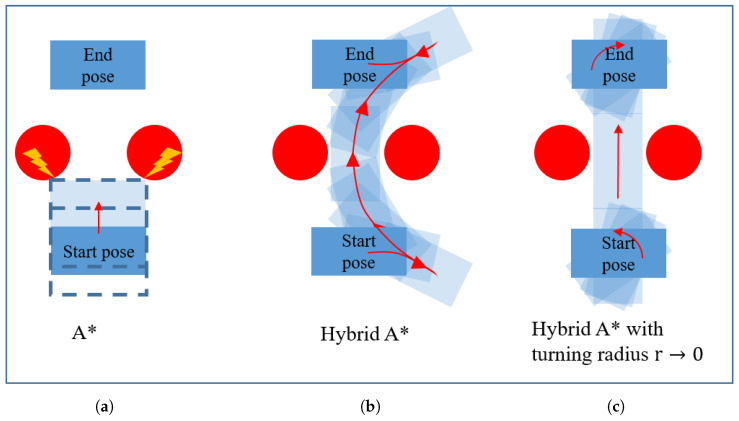
Path planning with different algorithms. (**a**) A* without the orientation, (**b**) Hybrid A* with greater turning radius and (**c**) turning radius approaching zero.

**Figure 16 sensors-25-04830-f016:**
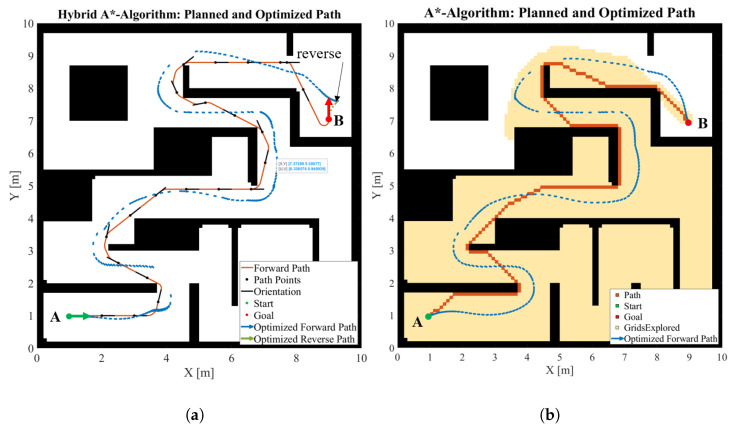
Comparison of the (**a**) Hybrid A* and (**b**) A* algorithm.

**Figure 17 sensors-25-04830-f017:**
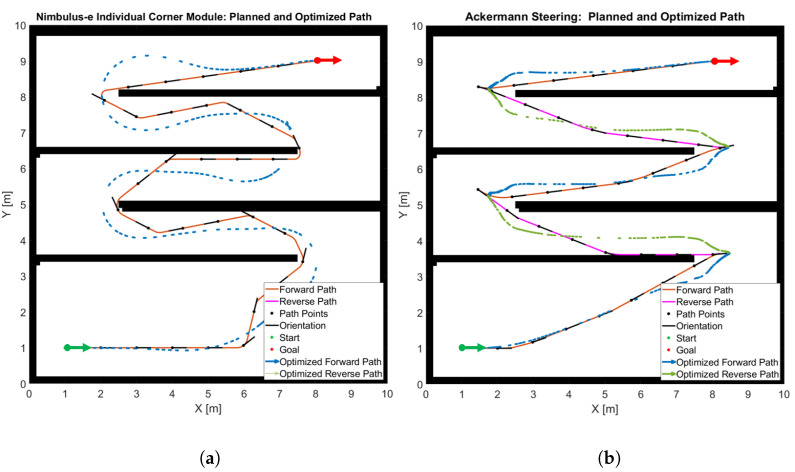
Maneuverability with (**a**) individual corner modules compared with (**b**) Ackermann steering.

**Figure 18 sensors-25-04830-f018:**
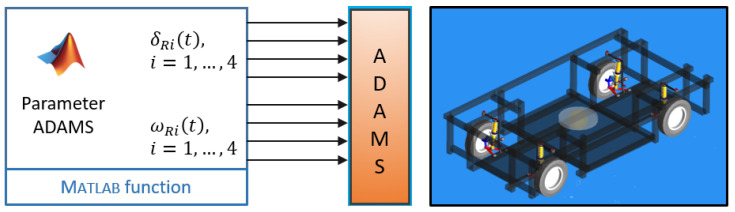
ADAMS CAR-MATLAB co-simulation.

**Figure 19 sensors-25-04830-f019:**
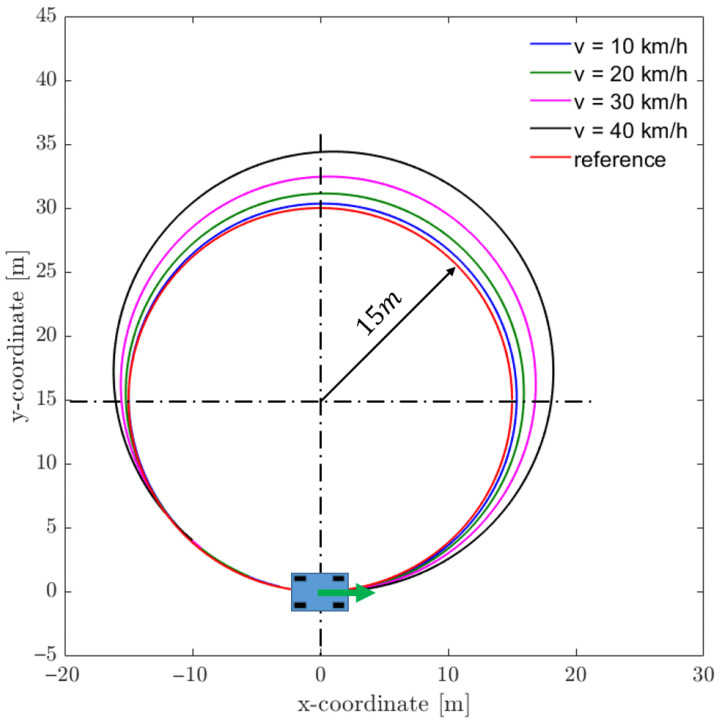
Circular target trajectory at different speeds.

**Figure 20 sensors-25-04830-f020:**
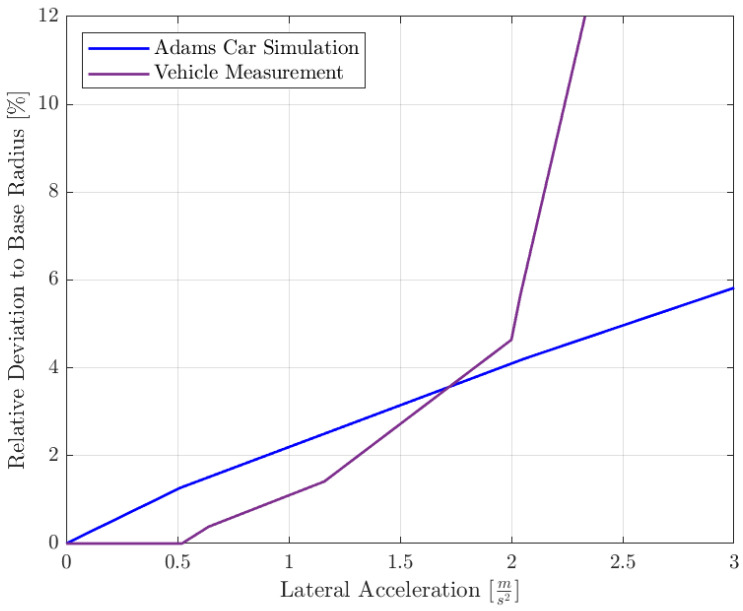
Correlation between simulation and physical test.

**Figure 21 sensors-25-04830-f021:**
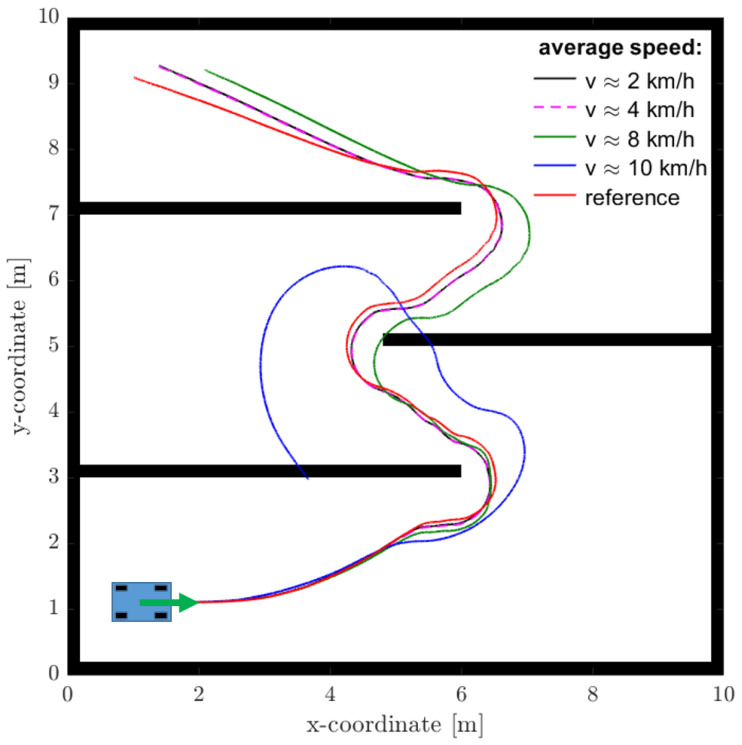
Path through a labyrinth at different speeds.

**Figure 22 sensors-25-04830-f022:**
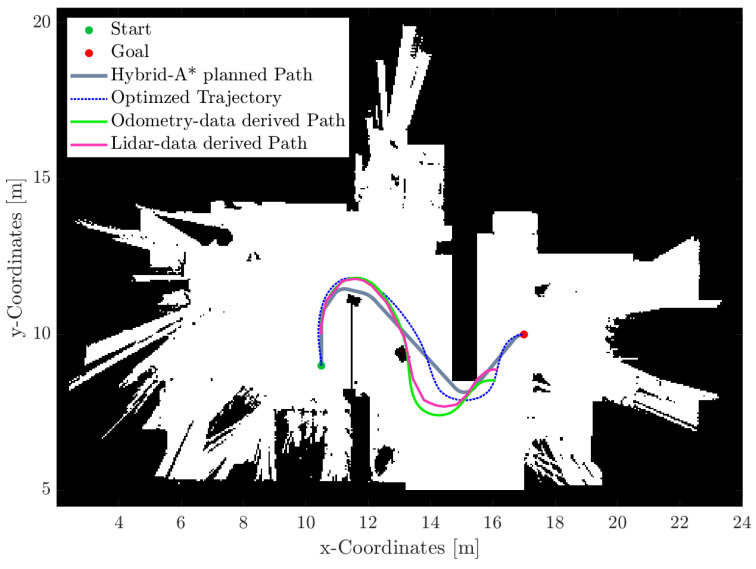
Comparison of the different trajectories.

**Table 1 sensors-25-04830-t001:** Comparison of RMSE, computational time, and computational effort for each method. Computational effort and time are not applicable to poses obtained from odometry as the calculations require a constant and very low number of arithmetic operations.

	Navigation Toolbox		LiDAR Toolbox		Poses from Odometry
	**With Poses**	**Without Poses**	**With Poses**	**Without Poses**	
RMSE	0.0453	0.0856	0.3692	0.2400	0.3648
CPU usage [%]	12.03	11.94	10.58	12.70	
Duration [s]	40.77	52.18	40.11	41.25	

## Data Availability

The datasets presented in this article are not readily available because the data are part of an ongoing study, and due to technical and time limitations. Requests to access the datasets should be directed to thomas.schmitz@thu.de.

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
