# Peer review of "Implementation of SLAM-Based Online Mapping and Autonomous Trajectory Execution in Software and Hardware on the Research Platform Nimbulus-e"

_sensors, 2025, doi:10.3390/s25154830_

Round 1

Reviewer 1 Report

Comments and Suggestions for Authors

In this paper studied the dynamic and implementation of a SLAM-based online mapping and autonomous trajectory execution system for the concept vehicle designed.
The experimental results obtained through investigation achieve and even surpass the design purpose. The paper has an original, scientific character.
I would like to remind you that the authors should make some corrections to the article.
1. Figure 6 should be expanded for references that should be included in the content - all other drawings have one, and the specificity of the presented system even requires that should be presented in this way.
The readability of the course shown in Figure 6 should be improved. In addition, the results shown in Figure 6 have been described too scarcely, it is recommended to extend the description of point 2.2.3. Electrical Architecture , so that the reader can obtain more information. 
2. SLAM is very common and does not require a long introduction. Instead, it is recommended to analyze the functional form of the error and the accuracy of SLAM in fitting such functions.
 3. Supplement the text of the manuscript (for example, introduction or conclusion) with specific results in the world and Europe, - insert links to the dissemination of results of other specialists - researchers who deal with similar issues. registered in the scopus/wos database with the title, such as: Development of a new system for attaching the wheels of the front axle in the cross-country vehicle, Industrial Robot Positioning Performance Measured on Inclined and Parallel Planes by Double Ballbar, Workshop programming as a part of technological preparation of production, Solutions to the characteristic equation for industrial robot's elliptic trajectories and Searching for collisions between mobile robot and environment, thanks.
3. figures 6, 13  and 18 should be contrasting and readable, 
4. conclusions and future work should be extended to contain practical applications based on research described in this paper, 
5. if possible highlight the course of dependencies/relations in figure No. 19 - the green color is indistinct , 
5. For ARTICLE type, 18 references are not enough. Please add more
references (>18) during your revisions, min. 25 references.
I recommend publishing the post after the proposed modifications.

Author Response

Dear reviewer,

Thomas

Reviewer 2 Report

Comments and Suggestions for Authors

This paper presents the design and implementation of an online mapping and trajectory planning/execution software system based on the Nimbulus-e hardware platform. Validation through ADAMS Car-MATLAB co-simulation and physical prototype testing demonstrates the system’s effectiveness. While the work makes a clear contribution by integrating SLAM into an innovative vehicle platform, the following issues require attention:

  1. Insufficient SLAM Performance Metrics (Fig. 12):Figure 12 compares trajectories from different SLAM methods but lacks quantitative performance indicators. Essential metrics such as pose error RMSE (vs. ground truth), loop closure detection success rate, and computational CPU usage should be provided. A comparative table summarizing positioning accuracy and computational efficiency is recommended.
  2. Inadequate Real-Vehicle Test Data:Physical prototype results are described qualitatively (e.g., "deviations increase with traveled distance"). Quantitative comparison with simulation results is missing. Provide statistical trajectory tracking error analysis (e.g., position/yaw error plots) during real-vehicle operation.
  3. Outdated Literature Review:The SLAM background cites Cadena (2016) and Barros (2022) but overlooks key advances in the last three years. Update references to include recent breakthroughs.
  4. Incomplete Figure Annotations:Figure 12: Axis units are missing (likely meters, but must be specified)ï¼›Figure 14: The basis for threshold selection in converting probabilistic to binary occupancy maps is unexplained. Justify the chosen threshold value

Author Response

Dear reviewer,

Thomas

Reviewer 3 Report

Comments and Suggestions for Authors

The paper presents the development and implementation of a SLAM-based online mapping and autonomous trajectory execution system for the Nimbulus-e, a nimble autonomous vehicle designed for confined spaces. The vehicle features individual steer-by-wire corner modules with in-wheel motors, enabling precise control via eight joint variables. A LiDAR sensor is used for real-time environment mapping and obstacle detection, with odometry data enhancing accuracy. The system employs A* and Hybrid A* algorithms for trajectory planning and optimization. Validation is performed through ADAMS Car-MATLAB co-simulation and a scaled physical prototype, demonstrating effective navigation in complex environments.

  1. The paper mentions the use of both A* and Hybrid A* algorithms for trajectory planning but does not clearly justify the choice of one over the other in specific scenarios.
  2. The integration of odometry data with SLAM is noted to reduce computation time and improve accuracy. However, the paper lacks quantitative results comparing SLAM performance with and without odometry.
  3. The prototype uses stepper motors with a worm gear ratio of 18:1, limiting steering resolution. How does this hardware constraint impact the vehicle's ability to follow highly precise trajectories?
  4. More attention should be paid to the discussion section, providing readers with more valuable insights.

Author Response

Dear reviewer,

Thomas
